# Generalized parton distribution functions of $\rho$ meson

**Baodong Sun**[1,2,3,4] **and Yubing Dong** [1,2,3]⋆

**1** Institute of High Energy Physics, The Chinese Academy of Sciences (CAS),
Beijing 100049, China
**2** School of Physical Sciences, University of Chinese Academy of Sciences,
Beijing 101408, China
**3** Theoretical Physics Center for Science Facilities (TPCSF), CAS, China
**4** Key Laboratory of Particle Physics and Particle Irradiation, Ministry of Education, Institute
of Frontier and Interdisciplinary Science, Shandong University, Shandong 266237, China

⋆ dongyb@ihep.ac.cn

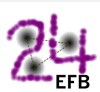 *Proceedings for the 24th edition of European Few Body Conference,*
## Abstract

We report our recent studies of generalized parton distribution functions of a $\rho$ meson with the help of a light-front constituent quark model. The electromagnetic form factors and structure functions of the system are discussed. Moreover, we show our results for its gravitational form factors (or energy-momentum tensor form factors) and other mechanical properties, like mass distributions, pressures, shear forces, and $D-$term.



## 1 Introduction

The study of generalized parton distribution functions (GPDs) is a key issue to understand the internal properties of a complex system [1–4]. Since the sum rules of GPDs relate to the form factors (FFs), and the GPDs in the forward limit connect to the parton distribution functions (PDFs), GPDs can give a three-dimensional description of the system. It is believed that the detailed information of GPDs can be obtained from deeply virtual Compton scattering or from the vector meson electro-productions.

There are many theoretical studies for the GPDs of nucleon (spin-1/2) [5–8], of pion (spin-0) [9,10], and of some nuclei (like $^3He$ [11], $^4He$ [12], and deuteron [13]). It should be mentioned that the experimental measurements for the Compton form factors of the nuclei [14–16]) have already been carried out in Jefferson Lab..

For spin-1 particles, like the $\rho$ meson and deuteron, there are also some discussions for their FFs, structure functions, transverse momentum distributions, PDFs, and GPDs in the literature [17–21]. It is addressed that the spin-1 particle, different from spin-1/2 and spin-0 ones, has three polarizations and therefore has a tensor structure function $b_1$, which relates to the parton distribution function of the longitudinally polarized target. There was an experimental measurement for the $b_1$ of the deuteron at HERMES [22], however, the available data cannot be simply understood by the deuteron structure functions constructed from the convolution approach by considering the deuteron being a weakly bound state of a proton and neutron [23]. It is expected that future Jefferson Lab. experiments would provide a more precise measurement of the deuteron tensor structure function.

We know that the vector meson $\rho$ is a spin-1 particle. It is believed as a two-body bound state with a quark and antiquark pair and its wave function is expected to be $S-$wave dominant. Since it is much easier to deal with the electromagnetic (EM) interaction to the quark than that to the nucleon, we focus our attention on the study of its GPDs.

## 2  Generalized parton distribution functions of a spin-1 particle

According to the general analyses of Ref. [13], there are nine parton helicity conserving GPDs for a spin-1 particle. In the quark sector, there are five unpolarized GPDs $H_i^q(x, \xi, t)$ (with $i = 1, 2, \cdots 5$ and the superscript $q$ standing for the quark contribution with flavor $q$), and four polarized GPDs $\tilde{H}_i^q(x, \xi, t)$ (with $i = 1, 2, \cdots 4$). Those GPDs are defined by the matrix elements of

$$
\begin{aligned}
V_{\lambda'\lambda} &= \frac{1}{2} \int \frac{d\kappa}{2\pi} e^{ix\kappa(P\cdot n)} < p', \lambda' \Big| \bar{\psi}\Big(-\frac{\kappa n}{2}\Big) \slashed{n} \psi\Big(\frac{\kappa n}{2}\Big) \Big| p, \lambda > \\
&= -(\epsilon'^* \cdot \epsilon) H_1^q + \frac{(\epsilon \cdot n)(\epsilon'^* \cdot P) + (\epsilon \cdot n)(\epsilon'^* \cdot P)}{P \cdot n} H_2^q - 2\frac{(\epsilon \cdot P)(\epsilon'^* \cdot P)}{M^2} H_3^q \\
&\quad + \frac{(\epsilon \cdot n)(\epsilon'^* \cdot P) - (\epsilon \cdot n)(\epsilon'^* \cdot P)}{P \cdot n} H_4^q + \Big[M^2 \frac{(\epsilon \cdot n)(\epsilon'^* \cdot n)}{(P \cdot n)^2} + \frac{1}{3}(\epsilon'^* \cdot \epsilon)\Big] H_5^q \\
&= \sum_{i=1}^{5} \epsilon'^{*\nu}(p', \lambda') V_{\nu\mu}^{(i)} \epsilon^\mu(p, \lambda) H_i^q(x, \xi, t),
\end{aligned}
\tag{1}
$$

for unpolarized case, and

$$
\begin{aligned}
\tilde{V}_{\lambda'\lambda} &= \frac{1}{2} \int \frac{d\kappa}{2\pi} e^{ix\kappa(P\cdot n)} < p', \lambda' \Big| \bar{\psi}\Big(-\frac{\kappa n}{2}\Big) \slashed{n} \gamma_5 \psi\Big(\frac{\kappa n}{2}\Big) \Big| p, \lambda > \\
&= -i\frac{\epsilon_{\mu\alpha\beta\gamma} n^\mu \epsilon'^{*\alpha} \epsilon^\beta P^\gamma}{P \cdot n} \tilde{H}_1^q + 2i\frac{\epsilon_{\mu\alpha\beta\gamma} n^\nu \Delta^\alpha P^\beta}{P \cdot n} \frac{\epsilon^\gamma(\epsilon'^* \cdot P) + \epsilon'^*(\epsilon \cdot P)}{M^2} \tilde{H}_2^q \\
&\quad + 2i\frac{\epsilon_{\mu\alpha\beta\gamma} n^\nu \Delta^\alpha P^\beta}{P \cdot n} \frac{\epsilon^\gamma(\epsilon'^* \cdot P) - \epsilon'^*(\epsilon \cdot P)}{M^2} \tilde{H}_3^q \\
&\quad + \frac{i}{2} \frac{\epsilon_{\mu\alpha\beta\gamma} n^\nu \Delta^\alpha P^\beta}{\mathcal{P} \cdot n} \frac{\epsilon^\gamma(\epsilon'^* \cdot n) + \epsilon'^*(\epsilon \cdot n)}{P \cdot n} \tilde{H}_4^q \\
&= \sum_{i=1}^{4} \epsilon'^{*\beta}(p', \lambda') \tilde{V}_{\beta\alpha}^{(i)} \epsilon^\alpha(p, \lambda) \tilde{H}_i(x, \xi, t),
\end{aligned}
\tag{2}
$$

for polarized case. In the above two equations, $\psi$ stands for the quark field, $M$ is the mass of the system, $\epsilon'(p', \lambda')$ (or $\epsilon(p, \lambda)$) is the polarization vector of the final (or initial) particle with

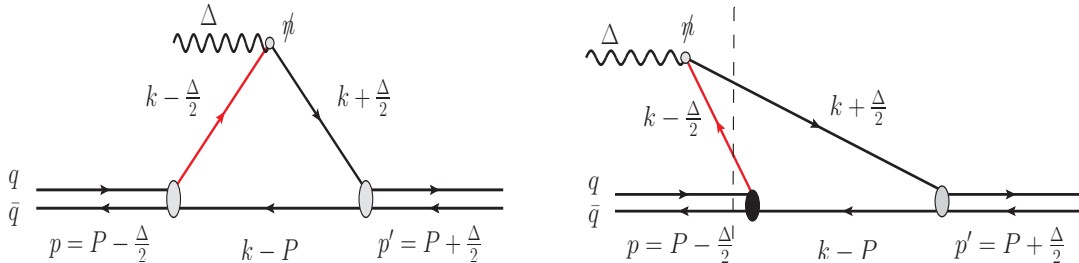

Figure 1: GPDs of the $\rho$ meson. (a) Direct Feynman diagram contributed to the GPDs by the stuck quark ($q$) in the valence region, and (b) the stuck $u$ quark in the non-valence region.

the momentum and polarization of $(p', \lambda')$ (or $(p, \lambda)$), respectively. In eqs. (1-2), $n$ is a light-like 4-vector with $n^2 = 0$, $P = (p' + p)/2$, $t = \Delta^2 = (p' - p)^2$. In addition, $\xi = -\frac{\Delta \cdot n}{2P \cdot n} = -\frac{\Delta^+}{2P^+}$, which is called skewness parameter describing the longitudinal momentum asymmetry. Figure 1 shows the GPDs of the $\rho$ meson with valence and non-valence contributions, respectively.

## 2.1 Form factors

We know that the sum rules of GPDs give the form factors of the system as

$$
\begin{aligned}
\int_{-1}^{1} dx\, H_i^q(x, \xi, t) &= G_i^q(t) \ (i = 1, 2, 3), & \int_{-1}^{1} dx\, H_i^q(x, \xi, t) = 0 \ (i = 4, 5); \\
\int_{-1}^{1} dx\, \tilde{H}_i^q(x, \xi, t) &= \tilde{G}_i^q(t) \ (i = 1, 2), & \int_{-1}^{1} dx\, \tilde{H}_i^q(x, \xi, t) = 0 \ (i = 3, 4),
\end{aligned}
\tag{3}
$$

where $G_{1,2,3}(t)$ are the known three form factors of the spin-1 particle which relate to the EM vector current

$$
I_{\lambda'\lambda}^{\mu} = \epsilon'^{*\beta} \epsilon^{\alpha} \left[ -2 \left( G_1(t) g_{\beta\alpha} - G_3(t) \frac{\Delta_{\beta} \Delta_{\alpha}}{2M^2} \right) P^{\mu} - G_2(t) \left( g_{\alpha}^{\mu} \Delta_{\beta} - g_{\beta}^{\mu} \Delta_{\alpha} \right) \right],
\tag{4}
$$

and they give the charge $G_C(t)$, magnetic $G_M(t)$, and quadrupole form factors $G_Q(t)$. $\tilde{G}_{1,2}(t)$ in eq. (3) are the two axial vector form factors defined by the electro-weak (EW) matrix element of

$$
\begin{aligned}
\tilde{I}_{\lambda'\lambda}^{\mu} &= <p', \lambda' | \bar{\psi}(0) \gamma^{\mu} \gamma_5 \psi(0) | p, \lambda> \\
&= -2i \epsilon^{\mu}{}_{\alpha\beta\gamma} \epsilon'^{*\alpha} \epsilon^{\beta} P^{\gamma} \tilde{G}_1(t) + 4i \epsilon^{\mu}{}_{\alpha\beta\gamma} \Delta^{\alpha} P^{\beta} \frac{\epsilon^{\gamma}(\epsilon'^{*} \cdot P) + \epsilon^{\gamma}(\epsilon' \cdot P)}{M^2} \tilde{G}_2(t).
\end{aligned}
\tag{5}
$$

It should be stressed that the form factors are only $t$-dependent and therefore, the sum rules in eq. (3) are $\xi$-independent although the GPDs of $H$ and $\tilde{H}$ are explicit $\xi$-dependent. This is due to the analytic properties of GPDs.

Furthermore, the energy-momentum tensor (EMT) $T^{\mu\nu}$ of the system relates to the gravi-

tational form factors (GFFs) as [24, 25]

$$
\begin{aligned}
< p', \lambda' | \hat{T}^{\mu\nu}(0) | p, \lambda > \; = \; & (P \cdot n) P^\nu \int x dx \int \frac{d\kappa}{2\pi} e^{ix\kappa(P \cdot n)} \bar{\psi}\left(-\frac{\kappa \cdot n}{2}\right) \gamma^\mu \psi\left(\frac{\kappa \cdot n}{2}\right) \qquad (6) \\
= \; & \Big\{ 2P^\mu P^\nu \Big(-\epsilon'^* \cdot \epsilon A_0(t) + \frac{\epsilon'^* \cdot P \epsilon \cdot P}{M^2} A_1(t)\Big) \\
& + \frac{1}{2}\big(\Delta^\mu \Delta^\nu - g^{\mu\nu}\Delta^2\big)\Big(\epsilon'^* \cdot \epsilon D_0(t) + \frac{\epsilon'^* \cdot P \epsilon \cdot P}{M^2} D_1(t)\Big) \\
& + 2\Big[P^\mu\big(\epsilon^\nu \epsilon'^* \cdot P + \epsilon'^{*\nu}\epsilon \cdot P\big) + P^\nu\big(\epsilon^\mu \epsilon'^* \cdot P + \epsilon'^{*\mu}\epsilon \cdot P\big)\Big]J(t) \\
& + \Big[\frac{1}{2}\big(\epsilon^\mu \epsilon'^{*\nu} + \epsilon^\nu \epsilon'^{*\mu}\big)\Delta^2 - \big(\epsilon'^{*\mu}\Delta^\nu + \epsilon'^{*\nu}\Delta^\mu\big)\epsilon \cdot P \\
& + \big(\epsilon^\mu \Delta^\nu + \epsilon^\nu \Delta^\mu\big)\epsilon'^* \cdot P - 4\epsilon \cdot P \epsilon'^* \cdot P g^{\mu\nu}\Big]E(t)\Big\} + ...,
\end{aligned}
$$

where $A_{0,1}(t)$, $D_{0,1}(t)$, $J(t)$, and $E(t)$ are the six energy-momentum conserved GFFs of the spin-1 system, and $\cdots$ denotes the other contributions from the energy-momentum non-conserved form factors. We know that GFFs can be extracted from the moments of GPDs [24,25]. Therefore, we can get the energy-momentum tensor as well as the mechanical properties of the system, like the mass distributions, shear forces, pressures, and the $D-$term.

For example, the mass radius is defined as

$$
< |r^2| >_{Grav.} = \frac{1}{M^2}\int d^3 r r^2 T^{00}(\vec{r}) = -6\frac{dA_0(t)}{dt}\Big|_{t\to 0}. \qquad (7)
$$

Then, the pressures and shear forces $p_i(r)$ and $s_i(r)$ $(i = 1, 2)$ are

$$
\begin{aligned}
\int \frac{d^3 \Delta}{2E(2\pi)^3} e^{-i\vec{\Delta}\cdot\vec{r}} < p', \lambda' | T^{ij}(0) | p, \lambda > = \; & p_0(r)\delta^{ij}\delta_{\lambda'\lambda} + s_0(r)Y_2^{ij}\delta_{\lambda'\lambda} \qquad (8) \\
& + p_2(r)\hat{Q}_{\lambda'\lambda}^{ij} + 2s_2(r)\big[\hat{Q}_{\lambda'\lambda}^{ip}Y_2^{pj} + \hat{Q}_{\lambda'\lambda}^{jp}Y_2^{pq} - \delta^{ij}\hat{Q}_{\lambda'\lambda}^{pq}Y_2^{pq}\big] + ...,
\end{aligned}
$$

where $Y_2^{ij} = \frac{r^i r^j}{r^2} - \frac{1}{3}\delta^{ij}$ and the quadrupole operator $\hat{Q}_{\lambda'\lambda}^{ij} = < p, \lambda' | \hat{Q}^{ij} | p, \lambda >$ with $\hat{Q}^{ij} = \frac{1}{2}\big(\hat{S}^i \hat{S}^j + \hat{S}^j \hat{S}^i - \frac{2}{3}S(S+1)\delta^{ij}\big)$. The appearances of $p_2$ and $s_2$ etc., in the above equation, are due to the fact that the spin-1 system has quadrupole form factor. In unpolarized case, i.e. under the average over the polarizations, $\hat{Q}_{\lambda'\lambda}^{ij} = 0$ and therefore only $p_0$ and $s_0$ survive. Finally, the $D-$term of the system is

$$
\begin{aligned}
D \; = \; & -\frac{2}{5}m\int d^3 r\big(r^2 Y_2^{ij}\big)\sum_{\lambda'\lambda}\frac{\delta_{\lambda'\lambda}}{3}T^{ij}(\vec{r}, \lambda', \lambda) = -\frac{4}{15}m\int d^3 r r^2 s_0(r) \qquad (9) \\
= \; & -D_0(0) + \frac{4}{3}E(0),
\end{aligned}
$$

which stands for the fundamental property of the system. A stable system requires a negative value for the $D$-term.

## 2.2 Parton distribution functions

In the forward limit, namely $\xi \to 0$, the parton distribution functions relate to GPDs

$$
\begin{aligned}
H_1(x, 0, 0) \; = \; & \frac{q^1(x) + q^{-1}(x) + q^0(x)}{3} = q(x) \to F_1(x), \\
H_5(x, 0, 0) \; = \; & q^0(x) - \frac{q^1(x) + q^{-1}(x)}{2} \to b_1(x), \\
\tilde{H}_1(x, 0, 0) \; = \; & q_\uparrow^1(x) - q_\downarrow^1(x) = \Delta q(x) \to g_1(x), \qquad (10)
\end{aligned}
$$

where $q^\lambda = q_\uparrow^\lambda + q_\downarrow^\lambda$ stands for the parton distribution with the polarization parallel ($\uparrow$) and anti-parallel ($\downarrow$) to the motion of the spin-1 particle with polarization of $\lambda$. $b_1(x)$ is the tensor structure function, which is unique for the spin-1 particle.

## 3 Numerical calculations

### 3.1 Light-front quark model

To describe the $\rho$ meson in the quark degrees of freedom, we follow Ref. [26] to write an effective Lagrangian for the meson-quark-quark coupling as

$$\mathcal{L}_{qq\rho} = -i\frac{m}{f_\rho}\bar{q}\Gamma^\mu\vec{\tau}q\cdot\vec{\rho}_\mu = -i\frac{m}{f_\rho}\Big[\bar{u}\Gamma^\mu u\rho_\mu^0 + \sqrt{2}\bar{u}\Gamma^\mu d\rho_\mu^+ + \sqrt{2}\bar{d}\Gamma^\mu u\rho_\mu^- + \bar{d}\Gamma^\mu d\rho_\mu^0\Big], \quad (11)$$

where $f_\rho$ is its decay constant, $m$ is the constituent quark mass. The phenomenological vertex $\Gamma^\mu$ reads

$$\Gamma^\mu = \left[\gamma^\mu - \frac{(2k - P - \frac{\Delta}{2})^\mu}{M_0 + 2m}\right]\Lambda\Big(k - \frac{1}{2}P, p\Big), \quad (12)$$

where the kinematic invariant masse $M_0$ is

$$M_0^2 = \frac{\kappa_\perp^2 + m^2}{1 - x'} + \frac{\kappa_\perp^2 + m^2}{x'}, \quad (13)$$

with $\kappa_\perp = (k - P)_\perp - \frac{x'}{2}\Delta_\perp$ and the LF momentum fractions $x' = -k_s^+/p^+ = (1 - x)/(1 - |\xi|)$. The phenomenological quark momentum distribution inside the $\rho$ meson is selected to be

$$\Lambda\Big(k - \frac{1}{2}P, p\Big) = \frac{c}{[(k - \frac{1}{2}P)^2 - M_R^2 + i\epsilon][(k - \frac{1}{2}\Delta)^2 - M_R^2 + i\epsilon]}, \quad (14)$$

where $M_R$ is the regulator mass and $k$ stands for the momentum of the active quark. This distribution represents the wave function of a bound state.

Then, we calculate the matrix elements of eqs. (1-2) and extract the GPDs of the system. In our phenomenological approach, we have three model-parameters, the quark mass, regulator mass $M_R$, and the constant $c$ in eq. (14). The last one can be determined by the normalization of the $\rho^+$ meson charge, and the former two parameters are optimally selected as $m = 0.403$ GeV and $M_R = 1.61$ GeV, respectively. After extracting GPDs, we can get the EM and EW form factors from the sum rules, the structure functions in the forward limit ($\xi \to 0$), the gravitational form factors, the energy-momentum tensor as well as other mechanical properties like the pressures, shear forces, and mass distributions.

In our calculation, we simultaneously consider the contributions from the valence and non-valence regions (see Fig. 1). In the non-forward limit, namely $\xi \neq 0$ and $|\xi| < 1/\sqrt{1 - 4M^2/t}$, the non-valence contribution is found to be sizeable, and we simply employ the prescription of Ref. [27] for the non-valence contribution. Our numerical results for GPDs show that we can reach the continuity from the valence to the non-valence regions, and moreover, the sum rules of eq. (3) are numerically preserved at different $\xi$ region.

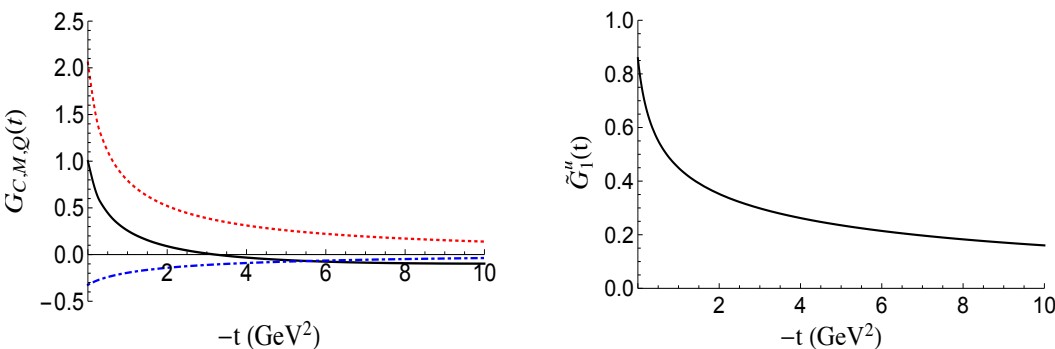

Figure 2: $\rho$ meson form factors. (a) EM form factors, charge $G_c$ (solid black curve), magnetic $G_M$ (dashed red curve), and quadrupole $G_Q$ (dotted-dashed blue curve) form factors, and (b) EW axial form factor $\tilde{G}_1^u(t)$ contributed by the $u$ quark.

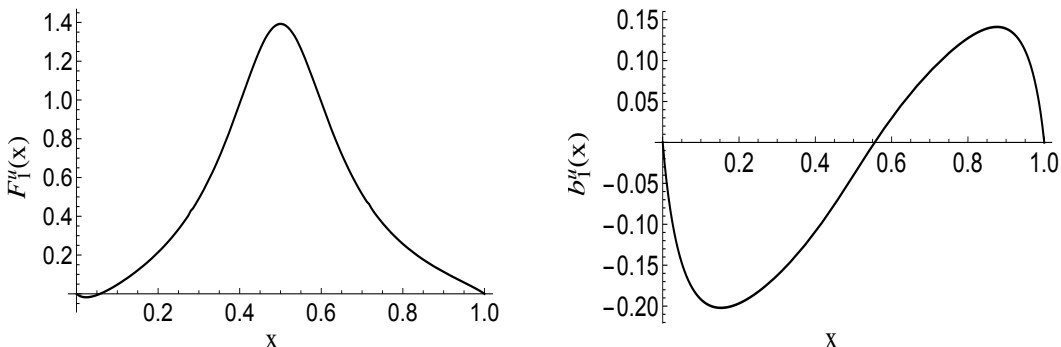

Figure 3: $\rho$ meson structure functions. (a) $F_1^u$ , and (b) $b_1^u(x)$ contributed by the $u$ quark, respectively.

### 3.2 Results of form factors

Our obtained "3-dimensional" GPDs have been explicitly plotted in Refs. [28, 29] at different skewness $\xi$, where the EM and EW form factors are also obtained according to eq. (3). Our calculated magnetic moment is $2.06/2M_\rho$, which agrees with other model calculations [19, 20]. The estimated quadrupole moment is $-0.323/M_\rho^2$ which is also consistent with other model calculations. In addition, our estimated charge radius is about 0.72 fm. Fig. 2 displays our calculated EM form factors and EW form factors $\tilde{G}_1(t)$ contributed by $u$ quark. Since we also calculate GPDs in the non-forward limit $\xi \neq 0$, by considering the contribution of the non-valence region, we check the sum rules and find that the sum of the contributions from the valence and non-valence regions at some values of $(\xi, t)$ is almost the same as the contribution from the valence region at the forward limit $(\xi = 0, t)$. Namely, our numerical results verify the sum rules of eq. (3).

### 3.3 Results of structure functions

In the forward limit $(\xi = 0)$, we get the structure functions from our GPDs. Fig. 3 show the results for $F_1^u(x)$ and $b_1^u(x)$. Since we employ the constituent quark model for the $\rho$ meson, the calculated structure functions are resulted from the constituent quarks. Our numerical result in Fig. 3 (b) implies that the known Close-Kumano [30] sum rule for the tensor structure function $\int dx\, b_1(x) = 0$ is almost preserved.

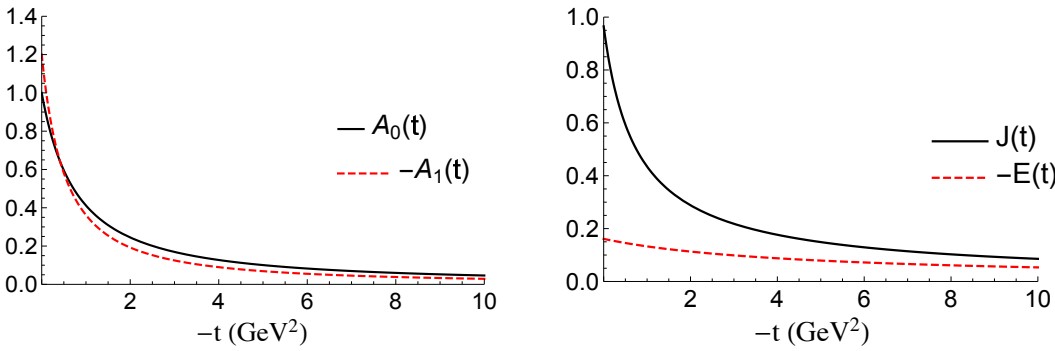

Figure 4: Some GFFs of the $\rho$ meson. (a) $A_0(t)$ and $A_1(t)$ , and (b) $J(t)$ and $E(t)$.

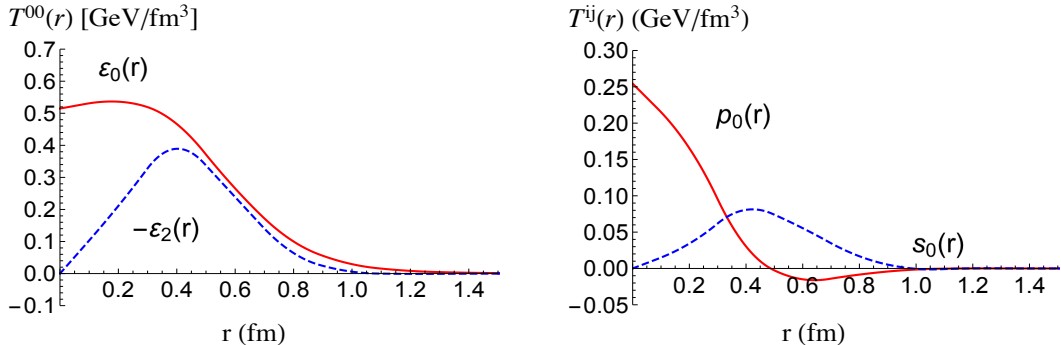

Figure 5: The mechanical properties of the $\rho$ meson. (a) mass distributions $\epsilon_0(r)$ and $\epsilon_2(r)$, and (b) shear force $s_0(r)$ and pressure $p_0(r)$.

### 3.4 Results of mechanical properties

From the moments of GPDs, we can get the gravitational form factors of the system. Fig. 4 shows four typical GFFs of $A_0(t)$, $A_1(t)$, $J(t)$, and $E(t)$, respectively. It should be reiterated that in the non-forward limit ($\xi \neq 0$), we also consider the non-valence contribution, and find that the sum of the valence and non-valence contributions to the GFFs are almost the same as the valence contribution in the forward limit ($\xi = 0$) at the same $t$.

Other mechanical properties of the system, like the mass distributions, pressures, shear forces, and $D-$ term of the $\rho$ meson can be calculated as well from the obtained GFFs and the energy-momentum tensors. Fig. 5 displays the results for the mass distributions, shear force, and pressure in our approach.

From eq. (7) we can get the mass radius. Our phenomenological approach gives $< |r^2| >_{Grav.}^{1/2} \sim 0.54$ fm, which is smaller than the calculated charge radius. This feature is consistent with the nucleon case [31, 32].

The pressure $p_0(r)$ in Fig. 5(b) is similar to the pressure obtained for the nucleon case as well [33]. Moreover, our calculated $D = -0.21$, which explicitly shows that the system is stable.

## 4 Summary

We summarize our recent studies for the properties of the $\rho$ meson (a typical spin-1 particle). We, first of all, calculate the GPDs of the system. Both the contributions from the valence

and non-valence regions are explicitly considered in the non-forward limit ($\xi \neq 0$). Our calculated low-energy observables, such as the form factors, are in a good agreement with other model and Lattice calculations. We also check the valence and non-valence contributions for the form factors and check the continuity from the valence to the non-valence regions. Our numerical results display that the obtained form factors are $\xi$-independent and the continuity preserves.

Then, by employing the forward limit, we obtain the structure functions, like $F_1(x)$, $g_1(x)$, and $b_1(x)$. The tensor structure function is unique for a spin-1 system. We find that our calculated $b_1$ almost satisfies the known Close-Kumano sum rule [30].

In particular, we calculate the moments of GPDs and extract the gravitational form factors. For the spin-1 system, it has six energy-momentum conserved gravitational form factors. The resulted GFFs give the energy-momentum tensor and the mechanical properties of the system, like mass distributions, shear forces, pressures, and the $D$-term. Our model calculation shows that the mass radius is about 0.54 fm which is smaller than its charge radius $\sim 0.72$ fm. The $D$-term $\sim -0.21$ represents that the considered $\rho$ meson is stable.

Finally, we know that the GPDs give a "3-dimensional" description for the space-like properties of the system. We can also calculate the time-like properties, like the generalized distribution amplitudes (GDAs) of the $\rho$ meson. In addition, we may further apply our approach to the deuteron target, which can be regarded as a weakly bound state of a proton and neutron.

## Acknowledgements

This work is supported by the National Natural Sciences Foundations of China under the grant Nos. 11975245 and 11475192, the Sino-German CRC 110 by NSFC under the grant No.11621131001, and the Key Research Program of CAS, under the grant No. Y7292610K1. Y. B. Dong thanks the support of Alexander Von Humboldt foundation and the hospitality of Institute of Theoretical Physics, Tuebingen University.

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
