# Peer review of "Generalized Parton Distribution Functions of $\rho$ meson"

_SciPost Physics Proceedings, doi:SciPost Phys. Proc. 3, 014 (2020)_

## Round 1 · Referee Report · Anonymous (Referee 1) · 2019-11-20

Report

This paper reports on calculations of generalized parton distribution functions, electromagnetic, electro-weak, and energy-momentum tensor form factors and other structure functions for the rho meson. The calculations are based on a phenomenological model for the rho-quark-antiquark vertex function in a constituent quark model. The momentum dependence of the vertex function is assumed to be of a particular functional form, depending on three parameters, one of which is fixed by a normalisation condition, whereas the other two are somehow "optimally selected" (no further details are given). It would clearly be more interesting to see results based on a dynamical calculation of the vertex function, but phenomenological results can still be useful for comparison purposes.

I think this paper merits publication. However, it should be improved in a few aspects:

  1. Equation (13) shows the momentum dependence $\Lambda(...)$ of the rho vertex, but the sentences before and after the equation do not explain what the exact structure of the vertex really is. Is it simply $\Gamma^\mu(...)=\gamma^\mu \Lambda(...)$, or are there other Lorentz structures included? Perhaps the sentence before (13), "... the phenomenological vertex $\Gamma^\mu$ equals to $\gamma^\mu$ plus the momentum-dependent term of the parton..." is misleading, and what is meant is "multiplied by" instead of "plus". This should be clarified.

My remaining comments concern the formatting and grammar of the paper.

Formatting:

  1. The figures 2-5 have really tiny labels, which makes them hard to read. If possible, they should be made bigger.

  2. All figures consist of a left and a right panel. In the captions, the authors refer to them as "((a), left)" and "((b), right)", respectively. This is quite unusual and rather clumsy, and it could be improved very easily: just give the left and right panels their labels (a) and (b), and then one can refer to them simply as "(a)" and "(b)" instead of "((a), left)", etc.

Grammar:

In general, the text is full of grammatical errors, which are too numerous to list them here (articles, prepositions, ...). Nevertheless, it is possible to understand most of what is written, so I will list here only the cases I find most important.

  1. Throughout the text, the authors write "share forces" instead of "shear forces".

  2. In the 3rd paragraph of the Introduction, "It is expected that future Jefferson Lab. would provide a more precise measurement..." should be changed to something like "It is expected that future Jefferson Lab experiments would provide a more precise measurement..."

  3. The decomposition of the energy-momentum tensor in Eq. (6) contains a mistake: the terms with $A_0(t)$ and $A_1(t)$ appear to be multiplied, but there should be some sign (probably a plus) between them.

  4. The sentence "which stands for the fundamental property and is negative characterizing a stable system." after Eq. (10) is confusing. Do they mean "which stands for a fundamental property..."?

  5. In the text before and after Eq. (13), the word "employ" is spelled "employe" twice.

I could go on with more examples, but perhaps the authors find somebody for a thorough proofreading of their text.

---

## Round 3 · Referee Report · Anonymous · 2019-12-20

Report
The authors have improved the manuscript in most aspects I mentioned in my report. Although there are still many small formal errors that could be eliminated, I think they are of minor importance and don't make the paper incomprehensible.
But I think that the second line of eq. (6) is still wrong, because the $A_0(t)$ and $A_1(t)$ GFFs are multiplied with each other (point 6 of my previous report). After correcting this mistake, I think the paper can be published.
Author: Yubing Dong on 2020-01-02 [id 693]
(in reply to Report 1 on 2019-12-20)Dear Editor,
Thanks for sending us the referee’s report on our manuscript and we very appreciate for our referees for their comments and suggestions. We polished our manuscript accordingly. Particularly, we corrected the typo of our eq. (6) .
We hope that the revised version of our manuscript is suitable for publication. Thanks again for your careful reading.
Best regards, Yubing Dong and Bao-Dong Sun

---

## Round 3 · Author Response

Thanks for sending us the referee’s report on our manuscript and we thank the referees for their comments and suggestions as well. We modified our manuscript accordingly. Our reply to the report is summarized as follows.
We hope that the present new version of our manuscript is suitable for publication. Thanks again for the comments.
Best regards,
Bao-Dong Sun and Yu-Bing Dong

---

## Round 3 · List of Changes

Referee’s report:
This paper reports on calculations of generalized parton distribution functions, electromagnetic, electro-weak, and energy-momentum tensor form factors and other structure functions for the rho meson. The calculations are based on a phenomenological model for the rho-quark-antiquark vertex function in a constituent quark model. The momentum dependence of the vertex function is assumed to be of a particular functional form, depending on three parameters, one of which is fixed by a normalisation condition, whereas the other two are somehow "optimally selected" (no further details are given). It would clearly be more interesting to see results based on a dynamical calculation of the vertex function, but phenomenological results can still be useful for comparison purposes.
I think this paper merits publication. However, it should be improved in a few aspects:
1. Equation (13) shows the momentum dependence Λ(...)Λ(...) of the rho vertex, but the sentences before and after the equation do not explain what the exact structure of the vertex really is. Is it simply Γμ(...)=γμΛ(...)Γμ(...)=γμΛ(...), or are there other Lorentz structures included?
Perhaps the sentence before (13), "... the phenomenological vertex ΓμΓμ equals to γμγμ plus the momentum-dependent term of the parton..." is misleading, and what is meant is "multiplied by" instead of "plus". This should be clarified.
Our reply:
We modified the part related to Sec. 3.1, and we clearly show what the vertex is employed in our approach. Please check the new manuscript.
REPORT:
My remaining comments concern the formatting and grammar of the paper.
Formatting:
2. The figures 2-5 have really tiny labels, which makes them hard to read. If possible, they should be made bigger.
3. All figures consist of a left and a right panel. In the captions, the authors refer to them as "((a), left)" and "((b), right)", respectively. This is quite unusual and rather clumsy, and it could be improved very easily: just give the left and right panels their labels (a) and (b), and then one can refer to them simply as "(a)" and "(b)" instead of "((a), left)", etc.
Our reply:
All the labels in Figures 2-5 are enlarged and the captions are simplified.
REPORT, Grammar:
In general, the text is full of grammatical errors, which are too numerous to list them here (articles, prepositions, …). Nevertheless, it is possible to understand most of what is written, so I will list here only the case I find most important.
4, Throughout the text, the authors write “share force” instead of “shear force”.
5, the 3rd paragraph of the introduction, “It is expected that future Jeffereon Lab. would provide a more precise measurement…”
should be changed to something like:
“It is expected that future Jeffereon Lab experiments would provide a more precise measurement…”
Our reply: We are sorry for the errors and modified the whole manuscript accordingly including the sentences, some articles, prepositions and grammar. We very appreciate for the careful reading of our referees, and hope the present version is more comprehensive to read.

---

## Editorial Decision

published